# FRUITBIN: A TUNABLE LARGE-SCALE DATASET FOR ADVANCING 6D POSE ESTIMATION IN FRUIT BIN PICKING AUTOMATION

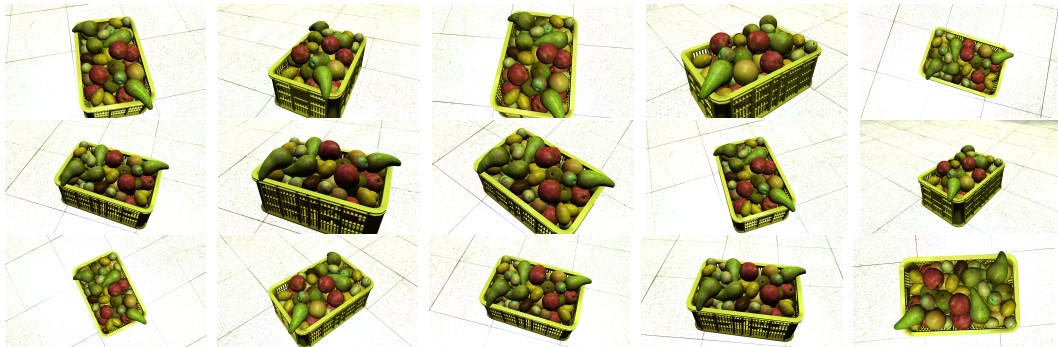

Figure 1: 15 camera viewpoints of a single scene from the dataset FruitBin.

## ABSTRACT

Bin picking is a ubiquitous application spanning across diverse industries, demanding automated solutions facilitated by robots. These automation systems hinge upon intricate components, including object instance-level segmentation and 6D pose estimation, which are pivotal for predicting future grasping and manipulation success. Contemporary computer vision approaches predominantly rely on deep learning methodologies and necessitate access to extensive instance-level datasets. However, prevailing datasets and benchmarks tend to be confined to simplified scenarios, such as those with singular objects on tables or low levels of object clustering. In this research, we introduce FruitBin. It emerges as an extensive collection of over a million images and 40 million instance-level 6D poses. Additionally, FruitBin differs from other datasets with its wide spectrum of challenges, encompassing symmetric and asymmetric fruits, objects with and without discernible texture, and diverse lighting conditions, all enriched with extended annotations and metadata. Leveraging the inherent challenges and the sheer scale of FruitBin, we highlight its potential as a versatile benchmarking tool that can be customized to suit various evaluation scenarios. To demonstrate this adaptability, we have created two distinct types of benchmarks: one centered on novel scene generalization and another focusing on novel camera viewpoint generalization. Both benchmark types offer four levels of occlusion to facilitate the study of occlusion robustness. Notably, our study showcases two baseline 6D pose estimation models, one utilizing RGB images and the other RGB-D data, across these eight distinct benchmarks. FruitBin emerges as a pioneering dataset distinguishing itself by seamlessly integrating with robotic software. That enables direct testing of trained models in dynamic grasping tasks for the purpose of robot learning. FruitBin promises to be a catalyst for advancing the field of robotics and automation, providing researchers and practitioners with a comprehensive resource to push the boundaries of 6D pose estimation in the context of fruit bin picking and beyond.

# 1 INTRODUCTION

Bin picking, a fundamental process where objects are retrieved from containers or bins, is widely utilized in various industries such as manufacturing, logistics, and warehousing. A notable application of this process is fruit bin picking, which poses unique challenges due to the diversity of fruit types and the potential for irreversible damage from incorrect vision-based grasping. This task is relevant in numerous fields, including agriculture, the food industry, and household robot assistance, where the availability of data is pivotal for further progress (Wang et al., 2022). The automation of fruit bin picking tasks is becoming increasingly important, yet it remains a complex topic due to the varying textures of objects, which limit the use of suction grasping. In this scenario, a more comprehensive understanding of the scene, such as 6D pose estimation, is required. State-of-the-art solutions in this area are data-driven and utilize robot perception techniques to perform object instance segmentation and estimate their 6D pose (Kleeberger et al., 2020; Grard et al., 2020). However, these solutions heavily depend on extensive datasets with diverse annotations at the object instance level, which can be prohibitively expensive to acquire in the real world.

Current benchmarks for 6D pose estimation predominantly emphasize computer vision aspects and lack integration within robotic learning software. Many benchmarks only offer partial robotic environments and overlook the crucial stage of seamless robot learning for manipulation, which involves mastering the intricate interactions between robots and objects (Dasari et al., 2019). Moreover, the majority of these benchmarks, including the popular Bop challenge (Sundermeyer et al., 2023), solely portray tabletop scenes featuring rigid objects, neglecting the specific challenge posed by bin-picking scenarios characterized by multiple object instances, significant occlusions, and clutter.

We introduce FruitBin, a comprehensive dataset consisting of simulated data tailored to facilitate robot learning, with a specific emphasis on the demanding task of fruit bin picking. Illustrative examples of FruitBin can be seen in Figure 1 or 3, and Table 1 offers a comparative overview of FruitBin in relation to state-of-the-art datasets. FruitBin is constructed upon PickSim (Duret et al., 2023), a recently introduced open-source simulation pipeline for robotics. PickSim provides dynamic configuration capabilities within Gazebo (Koenig & Howard, 2004), a widely adopted open-source 3D robotics simulation software extensively used in robotics research and development (Collins et al., 2021). The versatility of PickSim allows the vision model designed to be directly transferred and evaluated for dynamic tasks like grasping in the Gazebo simulator leveraging robotics frameworks such as ROS and Moveit.

The dataset comprises over 1 million images, along with a remarkable 40 million instance-level 6D pose annotations. It encompasses symmetric and asymmetric fruits, with and without texture, capturing the complexities all known 6D pose estimation challenges (Sahin et al., 2020; Sahin & Kim, 2019) within a single dataset, featuring varying viewpoints and lighting conditions across more than 70,000 scenes. FruitBin boasts comprehensive annotations and metadata, covering 6D pose, depth, segmentation masks, point clouds, 2D and 3D bounding boxes, and occlusion rates. The amalgamation of this extensive annotation set, its substantial scale, and its diversity of challenges positions FruitBin as an adaptable dataset for generating benchmarks.

To demonstrate its potential, we propose two distinct types of benchmarks for evaluating 6D pose estimation models, targeting new scene generalization and new camera viewpoint generalization. Each benchmark encompasses four levels of difficulty, incorporating occlusion scenarios. We evaluate the performance of two fundamental 6D pose estimation models, one employing RGB images and the other RGB-D images. To our best knowledge, FruitBin stands as the pioneering dataset meticulously tailored to address the demanding task of fruit bin picking (Wang et al., 2022). It represents the largest-scale dataset available for 6D pose estimation, offering robotic software compatibility and challenges that can be finely tuned to create custom benchmarks for 6D pose estimation challenges.

In the subsequent sections, Section 2 reviews prior work related to 6D pose datasets. Section 3 outlines the process of generating FruitBin using the PickSim software. Section 4 provides a description of FruitBin, including specific dataset statistics and sub-datasets tailored for the benchmark scenarios. The outcomes of the baseline 6D pose estimation models across the three scenarios are detailed in Section 5. Section 6 discusses certain limitations, while Section 7 concludes the article by providing insights and outlining future directions.

## 2 RELATED WORK

Data scarcity poses a significant hurdle for learning-based computer vision tasks, and this challenge becomes even more pronounced in the context of robot learning. This section provides a swift overview of the current state-of-the-art datasets for 6D pose estimation.

**6D pose datasets.** Numerous datasets have been established in the state-of-the-art for 6D pose estimation. Table 1 offers a comprehensive comparison of these datasets, encompassing various characteristics like data nature (real or synthetic), size (including the number of samples, scenes, and 6D pose annotations), and challenges presented (including occlusion, clutter, and multiple instances, among others). The proposed FruitBin dataset emerges for its comprehensive coverage of all current 6D pose estimation challenges within a single dataset. Significantly, FruitBin distinguishes itself by providing 2 to 1000 times more 6D pose samples and scaling the number of scenes from 6.4k to 70k. This enhancement in dataset size holds critical implications, especially in addressing the challenge of generalization to unknown scenes. Notably, FruitBin also encompasses additional annotations, such as occlusion rates, a feature exclusively shared with the MetaGraspNet dataset (Gilles et al., 2022). Furthermore, MetaGraspNet is unique in providing levels of difficulty annotations, underscoring the significance of such annotations. With over 40 million 6D pose annotations, FruitBin not only outperforms other datasets in terms of scale but also excels in the number of scenes and challenges covered, all while being seamlessly integrable within a robotic environment.

| Dataset | Data | #Samples | #Scenes | #6D pose | Challenges | O | C | Renv | Occ |
|---|---|---|---|---|---|---|---|---|---|
| LINEMOD Hinterstoisser et al. (2013) | R | 18k | 15 | 15k | TL | N | + | N | N |
| O-LINEMOD Brachmann et al. (2014) | R | 1214 | 15 | 120k | TL | + | + | N | N |
| APC Rennie et al. (2016) | R | 10k | 12 | ~240k | L | N | + | N | N |
| T-LESS Hodan et al. (2017) | R | 49k | 20 | 47k | TL+MI | + | + | N | N |
| YCB-V Xiang et al. (2018) | R-S | 133k | 92 | 613k | L | + | + | N | N |
| FAT Tremblay et al. (2018) | S | 60k | 3 | 205k | L | + | + | N | N |
| BIN-P Kleeberger et al. (2019) | R-S | 206k | 12 | 20M | MI+BP | +++ | +++ | N | N |
| ObjectSynth Hodan et al. (2019) | S | 600k | 6 | 21M | | + | + | N | N |
| HomebrewedDB Kaskman et al. (2019) | S | 17.4k | 13 | 56k | L | + | + | N | N |
| GraspNet-1B Maximilian et al. (2022) | R | 97k | 190 | 970k | - | ++ | ++ | N | N |
| RobotP Yuan et al. (2021) | S | 4k | - | - | TL | + | + | N | N |
| HOPE Tyree et al. (2022) | R | 2k | 50 | ~30k | MI+L+BP | + | + | N | N |
| MetaGraspNet Gilles et al. (2022) | R-S | 217k | 6.4k | 3M | MI+BP | ++ | ++ | Y | Y |
| SynPick Periyasamy et al. (2021) | S | 503k | 300 | 10M | BP | + | + | Y | N |
| StereOBJ-1M Liu et al. (2021) | R | 396k | 183 | 1.5M | L | + | N | N | N |
| DoPose Gouda et al. (2022) | R | 3k | 301 | 11k | BP | + | + | N | N |
| **FruitBin** | S | **1M** | **70k** | **40M** | **MI+BP+TL+L** | **+++** | **+++** | **Y** | **Y** |

Table 1: Comparison of 6D pose datasets with their diverse challenges (R: Real, S: Synthetic, O: Occlusion, C: Clutter, SO: Severe Occlusion, SC: Severe Clutter, MI: Multiple Instance, BP: Bin Picking, TL: Texture Less, L: Light variety). Occ indicates the availability of occlusion rates, and Renv indicates whether the dataset is integrable for application in a robotic environment.

## 3 RAW DATA GENERATION PROCESS OF FRUITBIN USING PICKSIM

This section outlines the process of generating FruitBin by harnessing the capabilities of Pick-Sim (Duret et al., 2023). PickSim, a recent pipeline, offers comprehensive annotation generation features, as illustrated in Figure 2, making it well-suited for applications in robot learning.

Creating task-specific datasets for robotics necessitates the development and training of tasks like vision and manipulation within the same environment. Employing robotic software, such as Gazebo, to generate synthetic computer vision data brings forth several advantages. Firstly, it allows for the seamless integration of physical engines, resulting in more lifelike outcomes. Secondly, it simplifies the integration of robots and sensors, equipped with native robot control capabilities. Lastly, it unleashes the potential to craft datasets tailor-made for robotic tasks, leveraging diverse open-source libraries like MoveIt (Coleman et al., 2014), integrated within Gazebo. PickSim further streamlines this process by providing user-friendly setup files for domain randomization, dataset recording, and

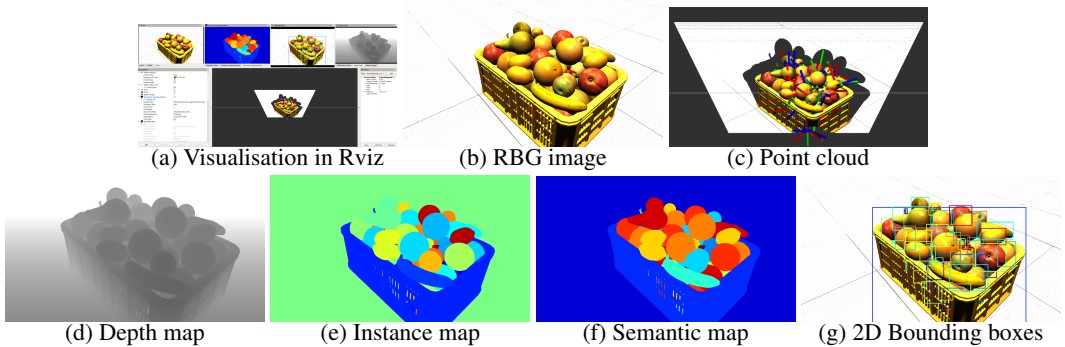

(a) Visualisation in Rviz        (b) RBG image        (c) Point cloud

(d) Depth map        (e) Instance map        (f) Semantic map        (g) 2D Bounding boxes

Figure 2: Examples of annotations generated with PickSim

generation. Each step of the pipeline can be effortlessly executed using straightforward commands, with parameters conveniently configured in JSON or YAML files. In this section, we present the four key steps involved in the generation of FruitBin using the PickSim pipeline.

**Pre-processing.**    For FruitBin, PickSim employs eight raw meshes representing one of the most common fruits to simulate: Apple, Apricot, Banana, Kiwi, Lemon, Orange, Peach, and Pear. To maintain the distinct characteristics of each fruit, no randomization is applied to the mesh or textures. Through this automated process, SDF files are generated, which are essential for Gazebo simulation. These SDF files contain crucial metadata, such as the category ID, necessary for future dataset recording.

**Scene randomization.**    PickSim (Duret et al., 2023) incorporates domain randomization techniques (Chen et al., 2022; Mishra et al., 2022; Muratore et al., 2021), utilized to generate diverse scenes for fruit bin picking. By using configuration files, users can easily customize object counts, cameras, and lighting conditions, eliminating the need for additional code and simplifying the creation of randomized Gazebo world files. In the FruitBin dataset, scene randomization encompasses the bin, fruits, and lighting. The bin undergoes randomization with rotations and color variations, while objects are subjected to position randomization atop the bin. The lighting setup includes randomized positions, intensities, and colors. This design ensures significant diversity in terms of lighting and overall scene composition. To maintain statistical consistency, the number of instances for each fruit category is randomly set between 0 and 30, ensuring a consistently full bin. Examples of these randomized scenes can be seen in Figure 3.

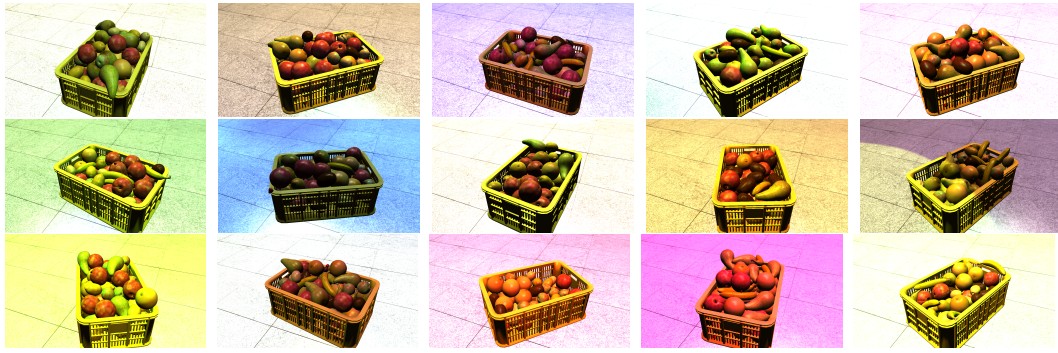

Figure 3: Initial 15 scenes from a single viewpoint illustrating the domain randomization.

**Camera randomization.**    The final facet of randomization involves camera settings, utilizing the orbiter sampler within PickSim to introduce variability in the distance (ranging from 0.55m to 1m) between the camera and the orbiter center, as well as varying angles to ensure optimal scene viewpoints. This seamless setup facilitates the generation of fully randomized scenes that are both physi-

cally realistic and well-suited for fruit bin-picking scenarios. The impact of these camera parameters is illustrated in Figure 1, which displays fifteen different viewpoints of a scene.

**Data recording.** Simulations in Gazebo can be effortlessly launched using the generated world files. These simulations yield datasets with recorded annotations, encompassing instance and semantic segmentation, bounding boxes, occlusion rates, 6D pose estimations, depth maps, point clouds, and normals. Although generation can be GPU-accelerated, the FruitBin dataset generation maximizes parallel processing using a CPU cluster. The estimated computational time for the whole generation process is approximately 40,000 CPU hours.

## 4 FRUITBIN: A LARGE SCALE DATASET

### 4.1 DATA AND STATISTICS

For FruitBin, which comprises eight different fruits, the randomization process is executed 10,000 times with 15 cameras, yielding 150,000 data frames. This process is repeated seven times. The aggregation of these seven parts forms the entirety of FruitBin, containing over 1 million frames across 70,000 scenes and 105 camera viewpoints. The dataset is meticulously organized, with metadata carefully stored. By segmenting the dataset into seven parts, sub-datasets can be formulated for scene generalization, camera generalization, and occlusion robustness. A thorough comparison of datasets is available in Section 2. In the provided code, a script is included to generate specific benchmarks for 6D pose estimation, incorporating user-defined parameters such as occlusion range, desired instance count, preferred viewpoints, and scene selections. While we have demonstrated this process with the example of eight benchmarks, users can create their own benchmarks tailored to their specific challenges.

Figure 4 provides an overview of data statistics and insights into the distribution of the multi-instance 6D poses among the various fruit categories. Remarkably, the complete scene randomization results in a Gaussian distribution of instance numbers in images, ensuring an equitable representation of each fruit category.

In order to enhance the versatility of our dataset, we've included supplementary data that emphasizes single-category bin picking. This type of data, where a bin contains only one type of fruit, is frequently encountered in real-world settings such as supermarkets and industrial environments. We also provide test data for both the general and single-category datasets, incorporating shape variations to broaden and assess their capacity to generalize to different shapes.

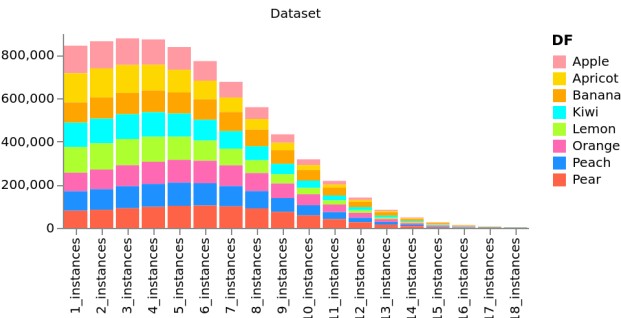

Figure 4: Statistics of the complete dataset for each category, indicating the instance count for each image.

### 4.2 A TUNABLE LARGE-SCALE DATASET FOR FRUIT BIN PICKING

Creating expansive, varied, and meticulously annotated datasets for 6D pose estimation is a demanding and time-intensive undertaking. The presence of thorough and well-annotated datasets holds immense significance in propelling advancements in 6D pose estimation. However, as illustrated in Table 1, each dataset introduces distinct challenges without mixing them. These challenges

encompass aspects like bin picking, scene diversity, viewpoint variety, diverse lighting conditions, occlusion, and multiple instances, among others. The FruitBin dataset, enriched with extensive annotations, a multitude of challenges, and substantial scale, provides exceptional tunability. The dataset's extensive scale permits the creation of sub-datasets customized for specific purposes or for executing ablation studies with provided scripts. The tunability feature of FruitBin is exemplified by its utilization in addressing two distinct types of 6D pose estimation benchmarks: scene generalization and camera viewpoint generalization, each encompassing four different levels of occlusion.

**Camera and scene generalization scenarios.** To investigate scene generalization and camera point-of-view generalization, we establish two distinct benchmarks for single-instance 6D pose estimation. The approach involves the careful sampling of the FruitBin dataset to generate scenario-specific datasets. The sampling process is performed using the initial portion of the dataset, encompassing 10,000 distinct scenes and 15 uniform camera viewpoints. As illustrated in figure 5, in the scene-oriented scenario, data is extracted from the extensive dataset to form training, evaluation, and testing subsets. Specifically, 60% of the samples, equivalent to 6,000 scenes with all 15 camera viewpoints, are allocated for training, while 20% are designated for evaluation, and the remaining 20% are reserved for testing, with each partition containing distinct scenes. A parallel methodology is applied to address the camera-oriented scenario. Here, nine initial viewpoints are assigned for training, three for evaluation, and the last three for testing. Throughout the dataset filtering process, all image samples are categorized based on their respective object categories, priming the data for future 6D pose estimation tasks. During the sampling process, additional metadata is captured, including scene IDs, camera IDs, category IDs, and the occlusion rate of the selected objects.

**Occlusion robustness scenarios.** To conduct a detailed examination of occlusion robustness, we have extended this analysis to the two aforementioned types of benchmarks. Specifically, we have incorporated four levels of difficulty related to occlusion, which leverage the available occlusion rate annotations. In the context of the previous sampling methodology, an occlusion parameter is introduced. Instead of utilizing the entire dataset, a filtering process is applied based on the occlusion rate associated with each object. The first version of the benchmark concentrates on objects with occlusion rates below 30%, followed by subsequent versions with occlusion rates of 50%, 70%, and 90%, respectively, progressively representing more challenging scenarios. During testing, further subdivision based on occlusion rate is executed to assess performance in occlusion-related scenarios. To delve even deeper into the study of occlusion impact, a partition within the testing phase could be established to provide an occlusion-aware performance analysis, as illustrated in Tables 6.

Figure 5 visually illustrates the data splitting in terms of image counts and the distribution among training, evaluation, and testing subsets for both types of benchmarks across the four levels of occlusion ranges.

## 5 EXPERIMENTS

**Baseline Methods.** To evaluate the challenges presented by the FruitBin dataset, we conducted an in-depth assessment using two distinct state-of-the-art 6D pose estimation models that utilize different data modalities.

The first method, known as **PVNet** (Peng et al., 2019), employs an RGB image and 3D model information of objects as input to predict the 6D pose. This approach consists of two stages: initially, it identifies the 2D keypoint locations of objects through a series of convolution and deconvolution blocks, followed by a RANSAC-based voting mechanism. Subsequently, the 6D pose is derived by solving an uncertainty-driven Perspective-n-Point (PnP) problem, utilizing the 2D keypoints and the 3D model. The second method, referred as **Densefusion** (Wang et al., 2019a), takes RGB image, depth information (RGB-D input), and a semantic mask of the scene as inputs. In the case of DenseFusion, it initially generates binary masks for each object, which are then employed to crop the image and point cloud within the region of interest (ROI). Each ROI serves as input to a 2D feature extractor and a point cloud extractor, leading to the acquisition of color and geometry embeddings. These embeddings are concatenated for each point and fused to generate 'local' and 'global' information, ultimately resulting in the dense fused features. The 6D pose is estimated via a pose predictor model that progressively refines the pose through iterative steps.

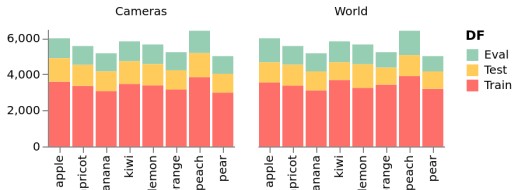
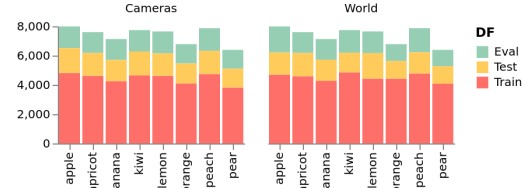

(a) statistics of the two types of benchmark taking into account an occlusion rate bellow 30%

(b) statistics of the two types of benchmark taking into account an occlusion rate bellow 50%

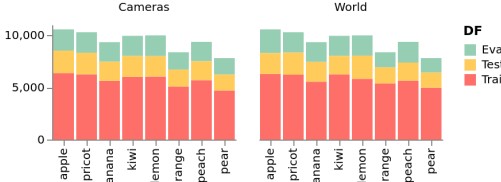
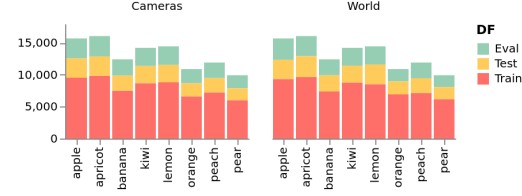

(c) statistics of the two types of benchmark taking into account an occlusion rate bellow 70%

(d) statistics of the two types of benchmark taking into account an occlusion rate bellow 90%

Figure 5: Statistical figures depict the image counts for each fruit category across the four occlusion ranges for both types of benchmarks (scene and camera generalization), further segmented by the train, evaluation, and testing partitions.

Both models have been trained and evaluated on established 6D pose estimation datasets: LINEMOD and YCB-Video, as discussed in Section 2. These models have demonstrated state-of-the-art results, confirming their suitability as representative baseline benchmarks.

**Metrics.** The baseline models are evaluated using the ADD (*average distance*) metric (Hinterstoisser et al., 2013) for non-symmetrical objects and ADD-S (*average closest point distance*) (Xiang et al., 2018) for symmetrical objects. In the case of FruitBin, apple, apricot, kiwi, lemon, orange, and peach objects are considered symmetrical, while banana and pear are non-symmetrical. In the following, ADD(-S) refers to both metrics. ADD is the mean distance of the transformed 3d model points using the estimated pose $[\hat{R}|\hat{t}]$ and ground truth pose $[R|t]$. Based on the computed distance, the estimated pose is considered correct if the distance is less than 10% of the model's diameter. The diameter represents the longest distance between 2 points in the object.

**Benchmark experiments.** Using the two baseline models described in Section 5 and the metric presented in Section 5, we trained each of them over the 8 benchmarks presented in Section 4.2. The results for Densefusion and PVNet are shown in Table 2.

Densefusion, which relies on depth information, exhibits, on average, superior performance compared to PVNet, which solely utilizes RGB images. It's important to note that, for specific studies on 6D pose estimation, Densefusion utilizes ground truth segmentation as input. However, the effectiveness of Densefusion is heavily influenced by object occlusion. It achieves a success rate of 90% or higher when the object's occlusion is below 30%. Conversely, as the occlusion in the data increases, up to 90%, the performance of Densefusion declines, reaching a success rate of 67%. It fails to meet the refinement threshold for this level of occlusion in the camera scenario. This observation is further substantiated by the results presented in Figure 6, which illustrate the performance across various occlusion ranges. Performance remains satisfactory when objects are slightly occluded, achieving a success rate of 90% with occlusion levels below 10%. However, the success rate drops significantly to 30% when considering occlusion levels between 80% and 90%.

On the contrary, PVNet exhibits distinct characteristics. The method relies on keypoints, which reduces its dependency on occlusion but shows less satisfactory results. It demonstrates consistent performance across different scenarios, yielding an average success rate of 64% (as shown in tables 2 and Figure 6). Notably, there are variations in performance based on the object, with notable peaks for pear, banana, and peach, which possess more intricate textures aiding the 2D feature extractor. Conversely, other objects in the dataset can be considered texture-less, lacking distinctive texture specifications. Another important aspect to consider is that PVNet relies on pixel values, while

| Occlusion range | Apple* | Apricot* | Banana | Kiwi* | Lemon* | Orange* | Peach* | Pear | Avg |
|---|---|---|---|---|---|---|---|---|---|
| **DenseFusion :** Benchmark scene generalisation | | | | | | | | | |
| From 0% to 30%: | 0.997 | 0.993 | 0.490 | 0.991 | 0.996 | 1.0 | 1.0 | 0.674 | 0.899 |
| From 0% to 50%: | 0.995 | 0.950 | 0.526 | 0.948 | 0.956 | 1.0 | 1.0 | 0.636 | 0.882 |
| From 0% to 70%: | 0.981 | 0.950 | 0.414 | 0.894 | 0.933 | 0.997 | 0.998 | 0.570 | 0.849 |
| From 0% to 90%: | 0.844 | 0.713 | 0.278 | 0.656 | 0.726 | 0.896 | 0.903 | 0.306 | 0.676 |
| **DenseFusion :** Benchmark camera generalisation | | | | | | | | | |
| From 0% to 30%: | 0.983 | 0.872 | 0.588 | 0.968 | 0.957 | 1.0 | 0.999 | 0.669 | 0.888 |
| From 0% to 50%: | 0.978 | 0.900 | 0.592 | 0.974 | 0.980 | 0.999 | 0.999 | 0.606 | 0.887 |
| From 0% to 70%: | 0.983 | 0.922 | 0.530 | 0.887 | 0.864 | 0.995 | 0.997 | 0.553 | 0.850 |
| **PVNet :** Benchmark scene generalisation | | | | | | | | | |
| From 0% to 30%: | 0.505 | 0.422 | 0.858 | 0.501 | 0.486 | 0.572 | 0.640 | 0.762 | 0.593 |
| From 0% to 50%: | 0.430 | 0.432 | 0.880 | 0.503 | 0.473 | 0.572 | 0.685 | 0.793 | 0.596 |
| From 0% to 70%: | 0.533 | 0.431 | 0.879 | 0.475 | 0.492 | 0.581 | 0.649 | 0.763 | 0.600 |
| From 0% to 90%: | 0.445 | 0.363 | 0.864 | 0.487 | 0.481 | 0.561 | 0.621 | 0.761 | 0.573 |
| **PVNet :** Benchmark camera generalisation | | | | | | | | | |
| From 0% to 30%: | 0.590 | 0.516 | 0.952 | 0.631 | 0.594 | 0.701 | 0.784 | 0.862 | 0.704 |
| From 0% to 50%: | 0.606 | 0.524 | 0.941 | 0.611 | 0.597 | 0.693 | 0.819 | 0.834 | 0.703 |
| From 0% to 70%: | 0.577 | 0.475 | 0.935 | 0.602 | 0.588 | 0.748 | 0.773 | 0.810 | 0.688 |
| From 0% to 90%: | 0.519 | 0.447 | 0.939 | 0.580 | 0.568 | 0.673 | 0.753 | 0.827 | 0.663 |

Table 2: Success rate results of the DenseFusion and PVNet models trained on the scene and camera benchmarks with different levels of occlusion are presented in the following table. The "Range of Occlusion" indicates the amount of occlusion in the data for the training, evaluation, and testing phases. Symmetric objects as marked with "*"

our dataset may exhibit varying distances between the camera and the objects. This can result in a limited number of pixels available for inferring keypoints and subsequently estimating the 6D pose.

These experiments highlight the significant advantages of our dataset and showcase the integration of the main challenges of 6D pose estimation into a single dataset. Indeed, on these benchmarks, the baseline methods do not consistently show satisfactory results across all fruit categories. It's important to note that these benchmarks encompass only two types of challenges. FruitBin provides ample room for increased difficulty, such as the addition of multi-instance 6D poses estimation challenges.

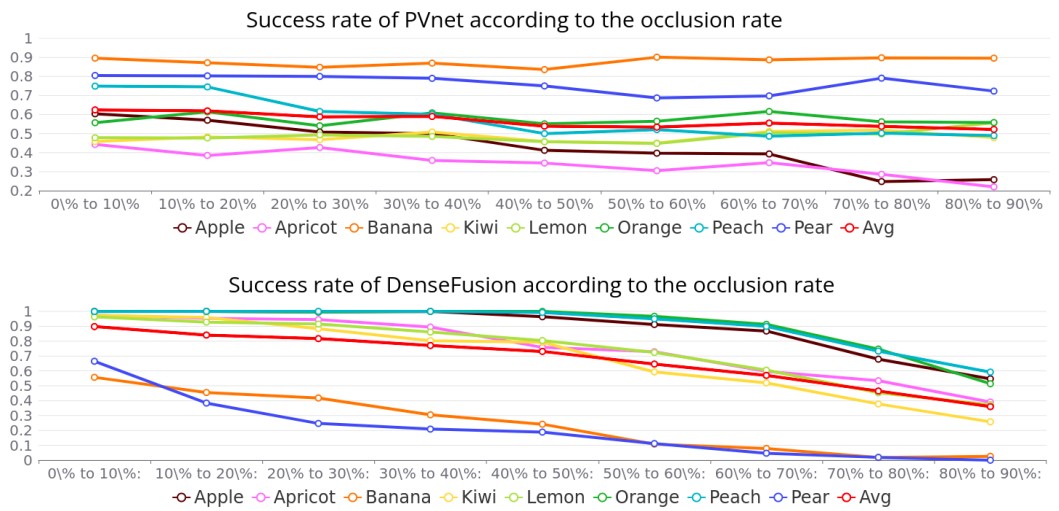

Figure 6: Precise evaluation of the DenseFusion and PVNet models (trained on the benchmark scene) is conducted using data with a range of occlusion from 0% to 90%, focusing on the testing partitions of different occlusion levels.

## 6 LIMITATION AND FUTURE WORK

This work aims to introduce a benchmark for 6D pose estimation by presenting a dataset specifically designed for this purpose. However, it is important to acknowledge that our dataset has limitations in terms of fruit meshes. Given the nature of fruits, where each instance is unique, there is a need to expand the dataset to include category 6D pose estimation. In order to address this, a logical step would be to incorporate new vision annotations into the open-source software PickSim NOCS (Wang et al., 2019b), which is widely utilized for category 6D pose estimation. This addition would enhance the dataset and enable complete evaluation and analysis of category 6D pose estimation methods.

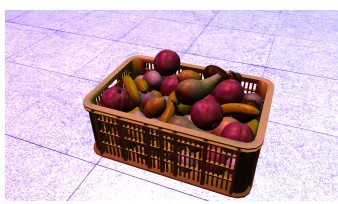 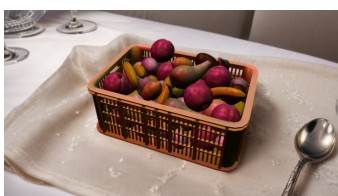 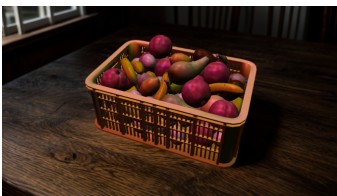

Figure 7: Left: the original image. Middle and right: two examples of new background generation using diffusion models.

Furthermore, addressing the sim2real gap between our simulator and real fruits is an important aspect to consider. We acknowledge the necessity for evaluating our domain randomization to address the sim2real gap. To alleviate this gap, we employ the latest advancements in diffusion models (bac) to substitute the image backgrounds. This strategy provides context to the images, significantly enhancing the sim2real gap bridging and facilitating a more varied image data augmentation. As a demonstration of this concept, we have included several examples of the generated images in Figure 7. To evaluate the practicality of our dataset in real-world scenarios, we plan to introduce two extensions. Firstly, we will supplement the existing dataset with a real-world testing dataset. Lastly, to fulfill the ultimate objective of this dataset, we will conduct a bin-picking grasping experiment on an actual robot. This will allow us to assess the performance of the 6D pose estimation.

## 7 CONCLUSION AND DISCUSSION

Introducing FruitBin, the largest dataset for fruit bin picking, featuring over 40 million 6D pose annotations and 1 million images. This dataset consolidates diverse challenges in 6D pose estimation, as showcased in Section 5. It addresses complexities like occlusion, symmetry, and low-texture objects, as examined in the dataset comparison in Section 2.

To address multiple facets of 6D pose estimation, we've devised eight benchmarks. These benchmarks evaluate scene and camera viewpoint generalization across four occlusion levels. While the current baseline model demonstrates its own merits, it's noteworthy that none of the baselines achieve satisfactory performance across all categories and benchmarks. This characteristic offers the research community an intricate challenge

Curated for 6D pose estimation, this dataset offers extended annotations for improved accessibility. Beyond its potential applications in 3D reconstruction, Nerf reconstruction, and multi-view 6D pose estimation, its value extends to robotics learning with its integration within the Gazebo simulator. The dataset bridges computer vision and robotics, fostering innovation. Researchers can assess models in simulations, advancing grasping and reinforcement learning. Our aim is for this dataset to catalyze the improvement of 6D pose estimation models and further the field of robotics learning.

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
