# OpenReview forum: "FruitBin: A tunable large-scale dataset for advancing 6D Pose estimation in fruit bin picking automation"
_ICLR.cc/2024/Conference — Submitted to ICLR 2024_

### Official Review · Reviewer_bGzb · 2023-10-23

**Soundness:** 3 good
**Presentation:** 3 good
**Contribution:** 2 fair
**Rating:** 5
**Confidence:** 5

**Summary:**

This paper introduces a large-scale PickSim-based synthetic dataset FruitBin for 6D object pose estimation in fruit bin picking. The dataset features comprehensive challenges and devised benchmarks for scene and camera view generalization as well as occlusion.

**Strengths:**

This is the first 6D object pose estimation dataset tailored for fruit bin picking although it is synthetic.

**Weaknesses:**

-- One drawback of Gazebo is that it can not do photorealistic rendering for objects and scenes with PBR textures. Although the generated dataset is large, without photorealistic textures, the transfer ability to real world is limited compared with other simulators such as BlenderProc and Kubric even the domain randomization techniques have been leveraged.

-- For real-world fruits, the size and shape of different instances of the same category vary to different degrees. However, it seems for FruitBin, these factors are not taken into consideration.

-- There is no real test set for the dataset, which is essential for sim2real and real-world applications.

-- The benchmarking methods are a bit outdated. PVNet and DenseFusion are from 2018-2019, but it is 2023 now.

-- It would be better to showcase some robotic applications like bin picking using this dataset, since it is targeted for fruit bin picking.

-- It would be better to mark symmetric objects with "*" in Table 2.

-- Table 1 is too wide.

-- There are some minor issues in the writing, more thorough proofreading is required.

**Questions:**

1) In the experiments, does PVNet use GT bboxes for cropping the objects in order to handle multiple instances of the same object class?

2) How does the diffusion generated backgrounds contribute to the performance?

---

> ### Author Response · Authors · 2023-11-22
> **Answer to the review**
>
> Dear Reviewer,
>
> We are grateful for your comprehensive review and constructive feedback on our manuscript. We have taken the time to address each of your comments and concerns as follows:
>
> **Photorealistic Rendering:**
>
> We concur with your observation regarding the limitations of Gazebo in terms of photorealistic rendering. While state-of-the-art generators such as Blenderproc and Kubric offer some advantages, they lack crucial occlusion annotation. Furthermore, we emphasize that models trained on our dataset can be integrated and tested for robotic simulation, a feature not guaranteed with Kubric and Blenderproc (we are not certain that the PBR rendering can be integrated on a camera in Pybullet, for example).
>
> We are cognizant of these limitations and believe that domain randomization is sufficient to reduce the sim2real gap. Moreover, as a general way of addressing the recurrent sim2real gap, data augmentation or domain adaptation [4] can further reduce the sim2real gap, and we demonstrated that a simple diffusion model can generate a good variety of realistic backgrounds.
>
> **Variation in Fruit Size and Shape:**
>
> While our current focus is on 6D pose estimation, we acknowledge the lack of variation in size and shape. Although not part of the benchmark yet, we leveraged PickSim to generate testing data where random scale variation over the three directions is applied to vary the shapes. Additionally, to answer real-world scenarios of industry and supermarkets, we generated bin picking for only one type of object. This comes with two variations: one where the mesh is unique and the second where mesh modification is applied.
>
> **Real Test Set:**
>
> We agree with your assertion on the importance of a real test set for sim2real and real-world applications. To this end, we intend to augment our dataset with real-world data to create a test set for evaluating sim2real performance.
>
> **Benchmarking Methods:**
>
> We concur that our benchmarks should include more recent methods. Although Pvnet and Densefusion are getting outperformed by recent methods, we believe that they remain representative baselines. It is worth noting that even with the recent result of the 6D pose BOP challenge 2023 [3], the difficulties are still the same. We can highlight non-negligible performance differences for occluded or cluttered datasets (LM-O, IC-BIN, ITODD) with scores below 0.8 and better than 0.9 for less occluded and cluttered ones ( T-LESS, YCB-V). This strengthens the need for occlusion studies in 6D pose estimation and benchmarks.
>
> We understand the importance of strong baselines in demonstrating the effectiveness of our approach, and **we are working to integrate GRD-NET**[1][2], which has been “The Best Open-Source Method” for the benchmarks in the BOP challenge in 2022 and 2023[3] to provide a more precise evaluation of our dataset.
>
> **Robotic Applications:**
>
> We appreciate your suggestion to demonstrate robotic applications such as bin picking using our dataset. For this rebuttal, we haven’t yet included grasping with the real robot. However, as an early proof of concept, a grasping simulation pipeline has been set up as follows:
>
> - we automatically generate a grasping list associated with the mesh,
> - we evaluate the 6D pose with our trained Densefusion model
> - we used free obstacle path planning libraries such MoveIt to grasp the object with the list of grasp and the 6D pose of the object
>
> **Symmetric Objects in Table 2:** We thank you for your suggestion and denoted symmetric objects with an asterisk (*) in Table 2 in the revised version of the paper.
>
> **Writing Issues:** We will address the writing issues you pointed out, including reducing the size of Table 1, separating additional work from the main paper, and improving the marking of symmetric objects in Table 2.
>
> **Questions:**
>
> **PVNet and GT Bboxes:** In all our generated benchmarks, we have only considered 1 instance of a category in the data. The filtering parameters for the benchmarks"
>
> **Diffusion-Generated Backgrounds:** The diffusion backgrounds have only been tested to showcase the feasibility of improving the sim2real gap. However, we haven’t explored how it would impact the model for 6D pose estimation.
> We hope that our responses address your concerns.
>
> [1]Wang, Gu et al. “GDR-Net: Geometry-Guided Direct Regression Network for Monocular 6D Object Pose Estimation.” 2021 IEEE/CVF Conference on Computer Vision and Pattern Recognition (CVPR) (2021): 16606-16616.
>
> [2] https://github.com/shanice-l/gdrnpp_bop2022
>
> [3] https://bop.felk.cvut.cz/challenges/bop-challenge-2023/
>
> [4] Truong, J., Chernova, S., & Batra, D. (2021). Bi-directional domain adaptation for sim2real transfer of embodied navigation agents. IEEE Robotics and Automation Letters, 6(2), 2634-2641.

---

### Official Review · Reviewer_gFnj · 2023-10-27

**Soundness:** 4 excellent
**Presentation:** 4 excellent
**Contribution:** 3 good
**Rating:** 8
**Confidence:** 5

**Summary:**

* This paper tackles novel research direction of fruits (or generalized any grocery item) using a robo-arm.
* Dataset uses RGB and depth cameras for curating and annotatings the dataset.

**Strengths:**

* This industry really needs a good dataset to further explore the problem, this paper just targeted that.
* This paper generalizes scenes as well as camera position for wider acceptability of it.
* Good reference to prior work on datasets.

**Weaknesses:**

* I would have preferred to see even more robust baselines.

**Questions:**

NA

---

> ### Author Response · Authors · 2023-11-22
> **Answer to the review**
>
> Dear Reviewer,
>
> Thank you for your review and positive feedback on our paper. We are pleased that you found our research direction and dataset valuable for the industry.
>
> We appreciate your suggestion regarding the need for more robust baselines. Although Pvnet and Densefusion are getting outperformed by recent methods, we believe that they remain representative baselines. It is worth noting that even with the recent results of the 6D pose BOP challenge 2023 [3], the difficulties are still the same. We can highlight non-negligible performance differences for occluded or cluttered datasets (LM-O, IC-BIN, ITODD) with scores below 0.8 and better than 0.9 for less occluded and cluttered ones (T-LESS, YCB-V). This strengthens the need for occlusion study in 6D pose estimation and benchmarks.
>
> We understand the importance of strong baselines in demonstrating the effectiveness of our approach. **We are working to integrate GRD-NET**[1][2], which has been “The Best Open-Source Method” for the benchmarks in the BOP challenge in 2022 and 2023[3], to provide a more precise evaluation of our dataset.
> Once again, we thank you for your time and constructive feedback. We look forward to incorporating your suggestions to improve our work.
>
> [1]Wang, Gu et al. “GDR-Net: Geometry-Guided Direct Regression Network for Monocular 6D Object Pose Estimation.” 2021 IEEE/CVF Conference on Computer Vision and Pattern Recognition (CVPR) (2021): 16606-16616.
>
> [2] https://github.com/shanice-l/gdrnpp_bop2022
>
> [3] https://bop.felk.cvut.cz/challenges/bop-challenge-2023/

---

### Official Review · Reviewer_J2Gb · 2023-10-29

**Soundness:** 2 fair
**Presentation:** 2 fair
**Contribution:** 1 poor
**Rating:** 3
**Confidence:** 4

**Summary:**

This paper presents FruitBin, a 6D pose estimation dataset for fruit bin picking with benchmarking over scene generalization, camera generalization and occlusion robustness. It contains over a million images and 40 million instance-level 6D poses.

**Strengths:**

- This paper proposes a large-scale dataset, which may facilitate future research for bin-picking tasks.
- The technical details are clearly presented.

**Weaknesses:**

- Limited Contribution
    - It seems that the technical contributions of this paper is just replacing the assets in PickSim with fruits. I don't think this contribution is sufficient for an ICLR paper.
    - All the data are collected in the simulator. It seems that no data is collected in the real world.
- Inconvient Platform
    - This paper uses ROS+Gazebo as its simulator platform, and claims it's for "seamless robot learning". However, I would think mujoco, PyBullet, or Isaac Gym are some more popular options in the robot learning community.
- Format Issues
    - Table 1 and the references are with format issues.
    - The supplementary materials should not be attached to the main paper.

**Questions:**

- Will the dataset include more samples collected in the real world?

---

> ### Author Response · Authors · 2023-11-22
> **Answer to the review**
>
> Dear Reviewer,
>
> We are grateful for the time and effort you have invested in reviewing our paper. We appreciate your feedback and would like to address your concerns.
>
> **Limited Contribution:**
>
> We acknowledge your concern regarding the technical contributions of our paper. However, we believe that our datasets, specifically designed for 6D pose estimation in bin picking, could significantly advance the development of 6D pose estimation models. Our unique dataset brings together major challenges in 6D pose estimation, and its large scale allows for the creation of specific benchmarks. This is not proposed by the current existing datasets. The recent 2023 BOP challenge results highlight that state-of-the-art 6D pose estimation models are still sensitive to complex scenes and occlusion, as evidenced by a drop of 0.1/0.2 for datasets with occlusion and bin picking, indicating room for improvement.
>
> Regarding the lack of real-world data, we agree that this is a significant limitation. We are currently collecting data from physical setups to enhance our dataset, allowing for a better evaluation of the sim2real gap and the performance of models under real-world conditions.
>
> **Inconvenient Platform:** We selected ROS+Gazebo due to its widespread adoption in the robotics community and its compatibility with various hardware. While Mujoco, PyBullet, and Isaac Gym are gaining popularity in the robot learning community, we maintain that ROS+Gazebo remains one of the most utilized platforms for robotic simulation [1]. To our knowledge, Mujoco, Pybullet, and Isaac Gym may be more low-level robot learning-oriented than Gazebo+ROS. The latter offers the advantages of a large community and integrates popular high-level libraries such as MoveIt. The default use of ROS enables users to easily apply their developed pipeline to real robots. Specifically to robot learning, even if maybe less popular, Gazebo is still a reliable choice [2,3].
>
> **Format Issues:** We apologize for the formatting issues in Table 1 and the references. These will be corrected in the revised version of the paper, along with the supplementary materials detached from the main paper.
>
> **Questions:** In response to your query, we do indeed plan to include samples collected in the real world in the dataset. We believe this enhancement will increase the dataset’s value for the community.
>
> [1] Collins, J., Chand, S., Vanderkop, A., & Howard, D. (2021). A review of physics simulators for robotic applications. IEEE Access, 9, 51416-51431.
>
> [2] Zamora, Iker, et al. "Extending the openai gym for robotics: a toolkit for reinforcement learning using ros and gazebo." arXiv preprint arXiv:1608.05742 (2016).
>
> [3] Ferigo, D., Traversaro, S., Metta, G., & Pucci, D. (2020, January). Gym-ignition: Reproducible robotic simulations for reinforcement learning. In 2020 IEEE/SICE International Symposium on System Integration (SII) (pp. 885-890). IEEE.

---

### Official Review · Reviewer_Ksan · 2023-10-30

**Soundness:** 2 fair
**Presentation:** 3 good
**Contribution:** 2 fair
**Rating:** 3
**Confidence:** 4

**Summary:**

The paper introduces a novel and extensive dataset designed for the task of fruit bin picking. This dataset is entirely synthetic and comprises 3D meshes of eight distinct fruits arranged in randomized configurations within bins, with varying lighting conditions and camera perspectives. The research employs this dataset to train two distinct models, one utilizing RGB data and the other incorporating RGB-D information, to serve as exemplary methods for 6-DOF pose estimation.

**Strengths:**

1. The paper is well-written and easy to understand. It explains its ideas clearly, making it accessible to a broad audience.
2. The dataset is extensive regarding images, configurations, and annotations.
3. The paper also offers detailed insights into the dataset, providing readers with a comprehensive understanding of its composition. This helps other researchers in utilizing the dataset effectively.
4. The synthetic nature of the dataset allows for the extraction of highly detailed annotations, which can be challenging to obtain in real-world scenarios.

**Weaknesses:**

1. One primary concern regarding the paper pertains to its real-world applicability. While the synthetic dataset's ability to provide detailed annotations is a strength, it also raises questions about the practical utility of algorithms trained on it in real-world scenarios. The paper should delve into the broader implications and limitations of applying such models to real-world fruit-picking scenarios.

2. A related concern is the limited variety of objects in the dataset. With only 8 types of fruits, and a significant majority of them being spherical (75%), the need for 6DOF pose estimation for these objects may be questionable. The paper should address the relevance of 6DOF pose estimation for objects that might not require such detailed positioning information.

3. The paper should explore the broader question of whether 6DOF pose estimation is necessary for fruit picking, particularly when considering that many real-world fruit-picking applications rely on suction grippers, making pose estimation less critical.

4. It is important to clarify the specific scenarios that the dataset targets. Random mixing of different fruits in bins may not represent common real-world scenarios, where fruits are typically harvested in monocultures and packed separately. The paper should outline the dataset's intended use cases and their alignment with real-world applications.

5. While the paper claims diversity in the dataset, I would argue that diversity should be measured by the variety of objects rather than the sheer number of images and annotations. The paper should address these concerns and clarify how the dataset's diversity aligns with its practical usefulness.

6. In my opinion, the representative images in the paper all look similar, and the lighting variations are synthetic without showing real-world visual phenomena (shadows, reflection). The paper should discuss how these factors affect the dataset's applicability to real-world scenarios and consider potential improvements.

7. Some of the language choices throughout the paper, such as the use of "comprehensive" to describe the evaluation using two models, are overly grandiose, in my opinion. The paper should adopt more precise and measured language to accurately represent the extent of the evaluation and avoid overstating its findings.

8. As this dataset targets robotic grasping of fruits, I would have liked to see a comparison of using the dataset on 6DOF grasping with a robotic gripper, not only pose estimation.

**Questions:**

1. In the intro, the paper mentions that the dataset contains delicate fruits like bananas and apricots that require haptic feedback for grasping, yet it is not mentioned how this is modeled and incorporated in the benchmark. Is this only in reference to exact pose estimation?
2. In Table 1. how does the presented dataset compare to other 6DOF datasets regarding object diversity?

---

> ### Author Response · Authors · 2023-11-22
> **Answer to the review - Part1**
>
> Dear Reviewer,
>
> Thank you for your insightful comments and constructive feedback. We appreciate the time and effort you’ve put into reviewing our paper. We agree with your concerns and have addressed them as follows:
>
> **Real-world applicability:** We acknowledge the limitations of synthetic datasets in replicating real-world conditions. Precise evaluation of our dataset for real fruits is an important question, given that we only have synthetic data. **We are currently scanning 3D real fruits and adding 6D pose real-world data to extend our dataset.** This dataset will be used for evaluating the sim2real gap of our dataset.
>
> **Variety of objects:** We agree that by targeting fruit bin picking, we inevitably limit the variety of shapes. Fruits are indeed mostly smooth compared to artificially created objects, such as industrial objects. However, fruit bin picking is a delicate task that requires careful handling to avoid damaging the fruits. We believe that knowing the semantics of the fruit and its position in the scene is of major importance during the grasping process to avoid any damage.
>
> **Necessity of 6DOF pose estimation:** We believe that 6DOF pose estimation isn’t incompatible with simple objects, and it can provide more precise control for robotic arms, even when using suction grippers. However, suction grasping requires fruits to be smooth, like apples, pears, or maybe bananas. In practical use, it will not be suitable for rugous textures such as lemons, oranges, or kiwis.
>
> **Intended use cases:** The dataset’s intended uses are multiple, with a goal of making it general for multiple purposes:
>
> - The first intended use is benchmark making as it gathers major challenges in 6D pose. It comes as a challenge for the community to make improvements in 6D pose estimation as it is commonly used with the popular BOP challenge.
> - In addition to these existing datasets for benchmarking, ours offers a useful and practical scenario that could easily occur in daily scenarios: fruit industry, house fruit bin, or even supermarket.
> - Mixing fruits in a bin is the most general and difficult scene we could create for 6D pose estimation purposes. However, it is right to note that real-world scenarios, such as industry or supermarkets, usually deal with only one category. In order to address this, we extended our dataset with data with bins of only one category of object.
>
> **Diversity of the dataset**: We understand your concern about the diversity of the dataset. We have included 8 fruits that we believe are the most common (apple, apricot, banana, kiwi, lemon, orange, peach, and pear). However, it remains in the range of the number of objects used by 6D pose datasets.
>
> **Lighting variations:** We agree that real-world visual phenomena like shadows and reflections are important. We are cognizant of these limitations and believe that domain randomization is sufficient to reduce the sim2real gap. Moreover, as a general way of addressing the recurrent sim2real gap, data augmentation or domain adaptation can further reduce the sim2real gap, and we demonstrated that a simple diffusion model could generate a good variety of realistic backgrounds and can create shadows or reflections.
>
> **Language choices:** We have revisited the language by removing some unnecessary adjectives.
>
> **Comparison with robotic gripper:** We appreciate your suggestion to demonstrate robotic applications such as bin picking using our dataset. For this rebuttal, we haven’t yet included grasping with the real robot. However, as an early proof of concept, a grasping simulation pipeline has been set up as follows:
>
> - We automatically generate a grasping list associated with the mesh.
> - We evaluate the 6D pose with our trained Densefusion model.
> - We used free obstacle path planning libraries such as MoveIt to grasp the object with the list of grasp and the 6D pose of the object.

---

> ### Author Response · Authors · 2023-11-22
> **Answer to the review - Part2**
>
> **Question:**
>
> In response to your query about modeling haptic feedback for delicate fruits, we currently do not incorporate this aspect into our dataset, as our primary focus is on pose estimation. We have clarified this point in the paper.
>
> Regarding object diversity, other benchmark datasets typically include between 8 (occluded linemod) and 39 objects (excluding GraspNet-1B, which primarily focuses on 6DOF grasping rather than 6D pose estimation). This is generally more than our dataset, as we focus solely on fruits. However, it’s important to note that even if the number of objects is low, datasets can pose a significant challenge for the 6D pose estimation community, particularly with occlusion. For instance, the popular dataset “occluded-Linemod” also has only 8 objects but remains one of the most challenging datasets. Indeed, in the recent BOP challenge 2023 [4], this dataset was one of the most difficult, with a score of 0.794 achieved by the winner of the challenge. This is compared to 0.928 for “YCB-video” which has 21 objects, or even “HomebrewedDB” with a score of 0.950 with 33 objects.
>
> We hope this addresses your concerns.
>
> [1] Zhongkui Wang, Shinichi Hirai, and Sadao Kawamura. Challenges and Opportunities in Robotic Food Handling: A Review, jan 2022. ISSN 22969144.
>
> [2] Tobin, Josh, et al. "Domain randomization for transferring deep neural networks from simulation to the real world." 2017 IEEE/RSJ international conference on intelligent robots and systems (IROS). IEEE, 2017.
>
> [3] Truong, J., Chernova, S., & Batra, D. (2021). Bi-directional domain adaptation for sim2real transfer of embodied navigation agents. IEEE Robotics and Automation Letters, 6(2), 2634-2641.
>
> [4] https://bop.felk.cvut.cz/challenges/bop-challenge-2023/

---

### Author Response · Authors · 2023-11-23
**General answer to the reviews - Part 1**

Dear Reviewers,

We sincerely appreciate your thoughtful review of our work and would like to address the valuable points you've raised regarding the limitations of our dataset. Your feedback is instrumental in enhancing the clarity and impact of our research.

**Sim2Real Gap:**

We concur with your observation that the Sim2Real gap is a crucial limitation of our study. To address this, we have conducted tests employing state-of-the-art **diffusion models to replace backgrounds**. These models have been widely acknowledged in image generation and processing. Surprisingly, in our dataset, they improve image realism significantly, even capturing intricate details like holes in the fruit bin and adapting to bin orientation. Leveraging these advanced diffusion models, in conjunction with precise prompt descriptions, offers unparalleled expressiveness that would be challenging to replicate purely through simulation. We believe that they can be utilized to substantially enhance the quality of our dataset.

We recognize the difficulties presented by the lack of precise real-world data for quantitatively evaluating the Sim2Real gap. To address this issue, we are **enhancing our dataset with real-world 6D poses**. This enrichment will facilitate a more thorough investigation of the Sim2Real gap and broaden the practical utility of our dataset.

**PickSim:**

Your insights regarding the choice of the PickSim simulator are deeply appreciated. While we acknowledge the limitations of realism inherent in Gazebo, it’s important to emphasize the strengths it brings to the dataset. Furthermore, PickSim includes **capturing diverse vision features such as segmentation maps, depth information, and occlusion rates**. All these features are often **absent in existing state-of-the-art generators such as Kubrik and Blenderproc**. By being generated by PickSim, Fruitbin offers a unique advantage not found in other datasets: **Its tight integration with the robotic simulator** allows for the seamless application of **trained 6D pose models to develop dynamic, vision-based grasping procedures without domain discrepancies**.

**FruitBin:**

We are grateful for your insights into the fruit application and the limitations concerning the number of categories in our dataset. The reasons for focusing on fruit bin picking are manifold. Firstly, our research centers around the manipulation of fragile and deformable objects, which includes fruits. Secondly, the lack of solutions in the domain of fruit bin picking and the absence of relevant datasets motivated our choice. Finally, we firmly believe that automating **fruit bin picking holds potential across industries, ranging from harvesting to industrial sorting and domestic robotic assistance**.

While the number of fruit categories could have been expanded our dataset includes most of representative shapes and textures such as bananas, pears, apples, and apricots. Although not part of the benchmark yet, we leveraged PickSim **to add testing data where random variation of scale over the three directions is applied to vary the shapes**. Additionally, to answer real-world scenarios of industry and supermarkets, **we generated bin picking for only one type of object. This comes with two variations: one where the mesh is unique and the second where mesh modification is applied.**

The key contributions of our work lie in the dataset's scale and its comprehensive coverage of 6D pose estimation challenges. The **provided script** empowers users to tailor benchmarks according to specific parameters, fostering research in point-of-view generalization, scene adaptation, and occlusion resilience. We emphasize that while the proposed challenges provide scope for benchmarking, researchers are free to **adapt them to their unique goals and objectives** related to 6D pose estimation.

**Robotic Applications:**

We appreciate your suggestion to demonstrate robotic applications such as bin picking using our dataset. For this rebuttal, we haven’t yet included grasping with the real robot. However, as an early proof of concept, a **grasping simulation pipeline has been set up** as follows:
- we automatically generate a grasping list associated with the mesh,
- we evaluate the 6D pose with our trained Densefusion model
- we used free obstacle path planning libraries such MoveIt to grasp the object with the list of grasp and the 6D pose of the object

---

> ### Author Response · Authors · 2023-11-23
> **General answer to the reviews - Part 2**
>
> **Paper :**
>
> We have **corrected numerous typos** throughout the paper. Moreover, we have streamlined Table 1 by eliminating the column that pertains to image resolution. We have also marked symmetric objects in Table 2 with the symbol ‘*’. We have **enhanced and clarified sections** of the paper, particularly those discussing **adding supplementary data**, the **necessity of 6D pose estimation for fruit bin picking*** and the **application of real robots**. We also mention the **creation of a real-world dataset**. The primary changes have been highlighted in red in the revised version of the paper.

---

### Meta-Review · Area_Chair_uaXa · 2023-12-06

**Metareview:**

This paper presents a dataset that is designed for the task of 6D pose estimation in fruit bin picking.  The dataset is synthetic with 8 types of fruits under varying lighting conditions and camera perspective. Two 6D pose estimation models utilizing RGB images or RGB-D data are evaluated on the proposed dataset.  The proposed dataset is large-scale and extensive regarding images, configurations, and annotation, which may be useful future research for fruit bin-picking tasks.  However, most reviewers consistently raised the concern regarding the limitation of the dataset because the dataset is synthetic only and has the limited variety of objects.  This concern raises questions about the practical utility of models trained on this dataset in the real-world scenarios.  The authors admitted this concern by addressing in their rebuttal that they are extending the dataset by adding 6D pose real-world fruit data.  The authors also acknowledged the lack of variation in size and shape.  The concern regarding the choice of the deployed benchmark methods was also raised. The rebuttal could not fully resolve this concern.  During the post-rebuttal discussion, it was acknowledged that a substantial revision followed by at least one more round of peer review is required to be a publishable state.  The paper should analyze the sim2read gap and offer more relevant benchmarks such as actual robot grasping of fruits.  The paper cannot be accepted, accordingly.

**Justification For Why Not Higher Score:**

N/A

**Justification For Why Not Lower Score:**

N/A

---

### Decision · Program_Chairs · 2024-01-16

Reject